# Predicting cell cycle stage from 3D single-cell nuclear-stained images

Gang Li[1,2,*], Eva K Nichols[1,*], Valentino E Browning[1]†, Nicolas J Longhi[1]†, Madison Sanchez-Forman[1], Conor K Camplisson[1], Brian J Beliveau[1,3], William Stafford Noble[1,4]

The cell cycle governs the proliferation of all eukaryotic cells. Profiling cell cycle dynamics is therefore central to basic and biomedical research. However, current approaches to cell cycle profiling involve complex interventions that may confound experimental interpretation. We developed CellCycleNet, a machine learning (ML) workflow, to simplify cell cycle staging from fluorescent microscopy data with minimal experimenter intervention and cost. CellCycleNet accurately predicts cell cycle phase using only a fluorescent nuclear stain (DAPI) in fixed interphase cells. Using the Fucci2a cell cycle reporter system as ground truth, we collected two benchmarking image datasets and trained 2D and 3D ML models—of support vector machine and deep neural network architecture—to classify nuclei in the G1 or S/G2 phases. Our results show that 3D CellCycleNet outperforms support vector machine models on each dataset. When trained on two image datasets simultaneously, CellCycleNet achieves the highest classification accuracy (AUROC of 0.94–0.95). Overall, we found that using 3D features, rather than 2D features alone, significantly improves classification performance for all model architectures. We released our image data, models, and software as a community resource.

## Introduction

The cell cycle plays a crucial role in regulating the growth, differentiation, and regeneration processes of all eukaryotic cells, ensuring proper cell function and maintenance throughout the organism's life. For this reason, profiling cell cycle dynamics is central to basic and biomedical research spanning development, health, aging, and disease. For example, cell cycle staging provides insights into cellular responses to drug therapies, immune cell activity during infection, metastatic potential of tumors, tissue regeneration, cellular differentiation, and more. Current methods for cell cycle staging often depend on metabolic labeling (e.g., with BrdU [1, 2]), genetic

engineering (such knock-in of fluorescently tagged proteins that oscillate with the cell cycle [3, 4, 5]), DNA staining for flow cytometry (propidium iodide, Hoechst, and/or DAPI [1, 6, 7, 8]), immunofluorescence against proliferative antigens (e.g., Ki67, PCNA [2, 9, 10, 11, 12]), or directly tracking by live-cell imaging. We reasoned that a machine learning (ML) approach may simplify cell cycle staging with minimal experimenter intervention and cost. Because a nuclear counterstain, such as DAPI, is already a part of many wet lab workflows—particularly those using fixed interphase cells—we aimed to test and develop ML-based models that are capable of classifying cell cycle stages based on DAPI-stained cell images alone.

Many methods have previously been developed for cell cycle staging across multiple data modalities, including single-cell HiC data [13, 14], single-cell RNA-sequencing [15, 16, 17], and microscopy data. In this work, we focused on cell cycle staging using microscopy image data. These methods are specialized for brightfield or phase contrast images [18, 19, 20, 21, 22 *Preprint*], light-sheet microscopy [23], or live-cell tracking [21, 24 *Preprint*, 25, 26, 27, 28, 29, 30 *Preprint*, 31]. We further narrowed our scope to include only studies that had similar input data and cell cycle staging tasks as this study: fixed interphase cells stained with DAPI and imaged by fluorescence microscopy (Table 1). Of the non-ML methods, Roukos et al [32] used two different methods to assign cell cycle stages to individual cells, based on three cell cycle stage bins (G1, S, and G2/M): either visually setting thresholds with respect to an empirical histogram of total DAPI intensity, or using commercial software (FCS express 5) to model the DAPI intensities with a Gaussian mixture model (GMM) [32]. Subsequently, Ferro et al developed a two-dimensional $k$-means clustering for cell cycle classification, based on the total DAPI intensity and the cell area [33]. Of the ML methods, Narotamo et al trained a support vector machine (SVM) to take into account these same two features, but they employed a two-label classification scheme (G1 versus S/G2) [34]. In separate work, the same authors also trained a type of deep learning model, a convolutional neural network (CNN), to infer cell cycle stages directly from DAPI-stained images, rather than from features inferred from those images [35].

---

[1]Department of Genome Sciences, University of Washington, Seattle, WA, USA  [2]eScience Institute, University of Washington, Seattle, WA, USA  [3]Brotman Baty Institute for Precision Medicine, Seattle, WA, USA  [4]Paul G. Allen School of Computer Science and Engineering, University of Washington, Seattle, WA, USA

Correspondence: beliveau@uw.edu; william-noble@uw.edu
*Gang Li and Eva K Nichols contributed equally to this work
†Valentino E Browning and Nicolas J Longhi contributed equally to this work

**Table 1.   Comparison of different cell cycle inference methods using DAPI-stained nuclei.**

| Method | Year | Output | Input | Dim | # of cells | Software Avail. | Data Avail. |
|---|---|---|---|---|---|---|---|
| GMM[32] | 2015 | G1, S, G2/M | DAPI intensity | 2D | Unclear | Commercial | Upon request |
| *k*-means[33] | 2017 | G1, S, G2/M | DAPI intensity and cell area | 2D | 10,805 | No | Upon request |
| CNN[35] | 2020 | G1, S/G2 | DAPI stained image | 2D | 3,553 | No | Upon request |
| SVM[34] | 2021 | G1, S/G2 | DAPI intensity and cell area | 2D | 3,553 | Yes[a] | Upon request |
| CellCycleNet | 2024 | G1, S/G2 | DAPI stained image | 3D | 7,802 | Yes | Yes |

[a]Code is available for the SVM, but not for processing images for input to the SVM.

One common feature of these existing methods is that they rely on 2D data: either 2D images or 2D maximum projections of 3D images. We hypothesized that a model that makes use of 3D images would be better equipped to accurately characterize the volume and surface area of the nucleus and hence would outperform a model that relies solely on 2D information. To test this hypothesis, we gathered in vitro microscopy data from fixed mouse fibroblast cells. To enable accurate labeling of cell cycle stages, we used the mouse fibroblast fluorescent ubiquitination-based cell cycle indicator ("Fucci2a") cell reporter line (5). This established Fucci2a cell line reports cell cycle stage through the fluorescent tagging of proteins that oscillate with the cell cycle: hCdt1, tagged with the mCherry fluorescent protein, and hGeminin, tagged with the mVenus fluorescent protein. The relative abundance of red and green signals within a cell's nucleus is tightly correlated with the cell cycle stage. The hCdt1 protein accumulates during G1, which degrades at the start of S phase. Meanwhile, hGeminin accumulates during S phase and is degraded at the start of G1 (5); G1 and G2/S phases of the cell cycle can therefore be faithfully resolved with the Fucci2a transgene. We then acquired comprehensive 3D image datasets of fixed Fucci2a cells across two common imaging modalities, epifluorescence and confocal microscopy. From these data, we trained a deep neural network model, CellCycleNet, to distinguish between nuclei in the G1 and S/G2 phases of the cell cycle. To have a straightforward training objective and to aid in comparisons to the prior art, we chose a binary classification model for testing.

Our results suggest that CellCycleNet provides more accurate cell cycle classification than the current state-of-the-art ML method, the SVM of Narotamo et al (34). Our fine-tuned CellCycleNet outperformed the SVM fitted to 2D features, achieving an area under the receiver operating characteristic curve (AUROC) of 0.95 on the epifluorescence data and 0.94 on the confocal data, representing an improvement of 7–8%. Furthermore, to verify the benefit of using 3D features, we directly compared two SVM models that rely upon the same input features: one model derives its features from 2D maximum projections whereas the other derives its features directly from 3D images. In an independent test set, we observed that the use of 3D features boosts the performance of the SVM model by 2–4%.

One challenge that this field faces is a lack of standardized benchmarks and publicly available data and/or software. Of the four references highlighted in Table 1, either the associated ground-truth cell cycle labels were not readily available, the image data was not sampled enough to be truly 3D, and/or our queries to the authors were not answered. These factors prevented us from readily using currently existing image data for testing CellCycleNet. To address this problem, we make our data, software, and associated benchmarking scripts publicly available. In particular, we provide CellCycleNet as open-source software, which may be used out-of-the-box or fine-tuned as needed. CellCycleNet is implemented in Python and is available with an MIT license at https://github.com/Noble-Lab/CellCycleNet. We anticipate that our trained CellCycleNet model will be generally applicable in any in vitro setting requiring cell cycle staging and that our benchmarking platform will be useful for future comparisons of cell cycle staging techniques.

# Results

## Generating benchmark 3D image datasets from two image modalities

Widefield epifluorescence microscopy and confocal microscopy are among the most common fluorescence-based imaging modalities in laboratories. We reasoned that imaging Fucci2a cells across these two platforms would provide a comprehensive training and testing dataset for use in developing ML models. Accordingly, we collected datasets on two instruments available to us that had suitable hardware configurations for the high-throughput acquisition of the Fucci2a cells (5) at sub-cellular and sub-nuclear resolution to facilitate robust feature extraction and ground-truth label assignment. As these systems have distinct optical configurations (e.g., widefield epifluorescence versus spinning disc confocal), there were differences in the lateral and axial sampling rates used to acquire the two datasets (see the Materials and Methods section). The epifluorescence dataset voxel dimensions are 188 nm in XY and 500 nm in Z. The confocal dataset's voxel dimensions are 183 nm in XY and 200 nm in Z. We used Cellpose for 3D nuclear segmentation (36, 37, 38 *Preprint*) (see the Materials and Methods section).

We collected in total 6,390 images containing 5,331 nuclei for the epifluorescence image dataset and 93,750 images containing 5,638 nuclei for the confocal image dataset. To ensure that only high-quality cells were used in model training, we removed improperly segmented cells (5.7% of cells in the epifluorescence dataset and 4% in the confocal dataset) or cells that did not express the Fucci2a transgene (17.8% of cells in the epifluorescence dataset and 29.9% of cells in the confocal dataset). Improper segmentations include doublet cells, cell debris, and truncated cells at tile borders. About

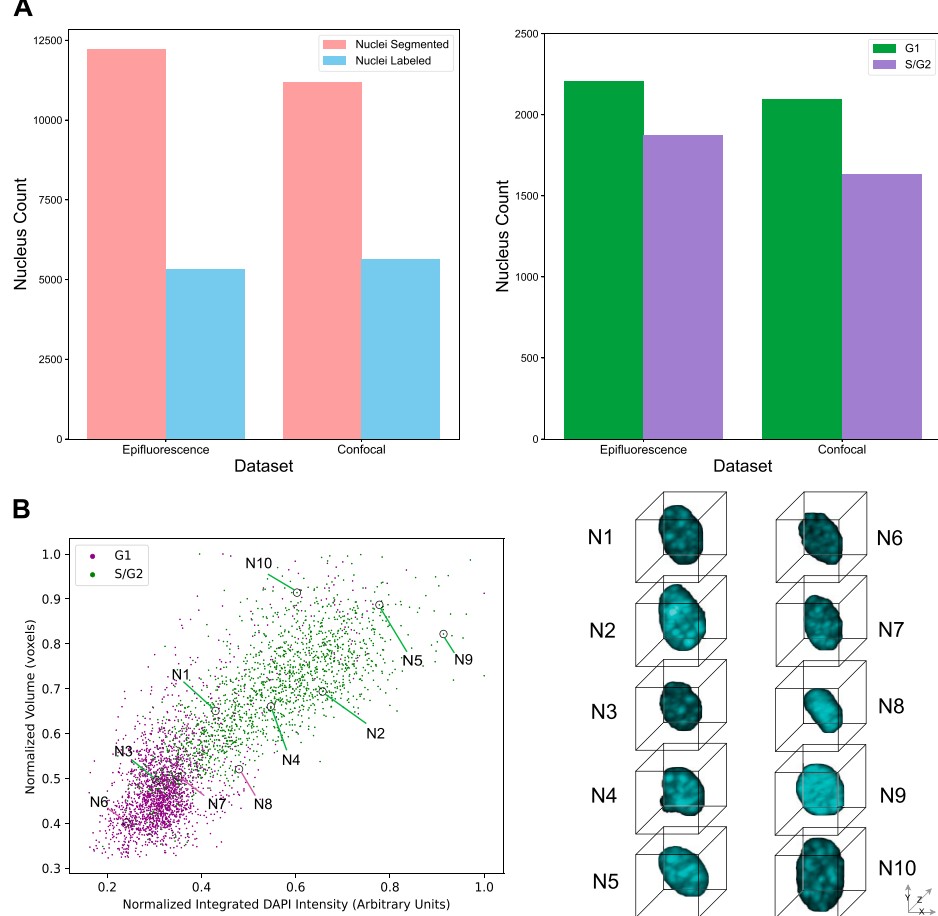

**Figure 1. Overview of image-based cell cycle dataset generation and ground truth labeling.** **(A)** Two new benchmarking 3D image datasets. **(B)** Scatter plots of confocal image datasets with top 10 representative cell images.

65–75% of the cells in each dataset passed quality control. After quantifying the relative mCherry- and mVenus-boosted fluorescent intensity per cell, we omitted cells of indistinguishable class along the S-phase border, due to the limited resolution power of the Fucci2a transgene. Finally, we assigned ground-truth labels, following Narotamo et al (2021) (34), to 4,076 cells (G1: 2,203 cells; S/G2: 1,873 cells) in the epifluorescence dataset and 3,726 cells (G1: 2,096; S/G2: 1,630) in the confocal dataset (Fig 1A). Scatter plots of cells from two datasets and images of 10 representative cells show that larger and brighter cells are more likely to be in S/G2 (Figs 1B and S1). The representative cells are selected by Apricot, which uses submodular optimization to summarize large data sets into minimally redundant subsets that are representative of the original data (39).

## CellCycleNet provides accurate cell cycle inference

We hypothesized that a deep neural network trained from our benchmark dataset could accurately assign cell cycle stage labels to individual cells. One lesson from the many recent successes of deep learning in various fields is that highly accurate models are often parameter-rich and trained from massive amounts of data. Although our data set is large—545 gigabytes of data—it is still small in comparison to image compendia used to train state-of-the-art image processing models. For example, ImageNet (40), a widely used dataset in image recognition, includes over 14 million images (last updated in March 2021), serving as a benchmark for training and evaluating model(s). Similarly, TissueNet (41), a dataset for segmentation models, contains over 1 million manually labeled cells, which is an order of magnitude larger than any previously published segmentation training dataset. Accordingly, we trained our model, called "CellCycleNet," using a transfer learning setup, which means that we used as a starting point a model that was previously trained from a very large collection of data to solve a related image analysis task. More specifically, we sought to fine-tune a pre-trained neural network that was designed to segment nuclei in light-sheet microscopy images (42). The model is a 3D CNN that processes 3D images through various operations, including multiple layers of convolution and max pooling. This transfer learning step helped to accelerate convergence. The final output is a probability score indicating whether the input cell image belongs to the G1 phase or the S/G2 phase. The network consists of 35 layers and has a total of 1,757,267 parameters (Figs 2A and S2). We divided each image dataset into a 70/20/10 split for training, validation, and testing, respectively. For optimization, we employed binary cross-entropy loss and the Adam optimizer.

We chose the SVM from Narotamo et al (34) as the state-of-the-art method because it was the most recent ML method for this task with similar input data (Table 1). Though Narotamo et al have also

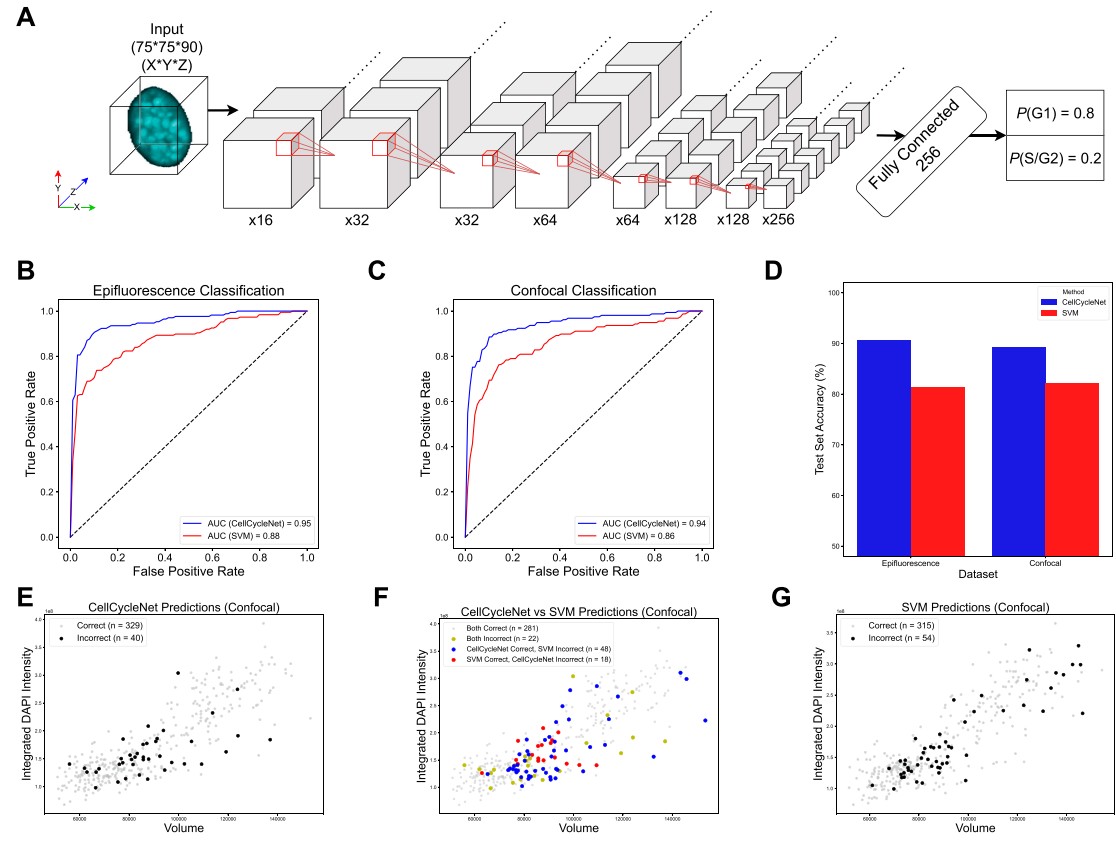

**Figure 2. CellCycleNet improves cell cycle inference on both benchmarking datasets.**
**(A)** The schematics of CellCycleNet. DAPI blue intensity is used as input, and ground-truth labels are generated using immunofluorescence-amplified hCdt1 (mCherry; also referred to as RFP) and hGeminin (mVenus; also referred to as GFP) signal. **(B)** ROC curves on epifluorescence data. **(C)** ROC curves on confocal data. **(D)** Bar plot of test accuracy. **(E)** Scatter plot of CellCycleNet predictions. **(F)** Scatter plot of prediction differences between two models. **(G)** Scatter plot of SVM predictions.

developed a 2D CNN in older work (35), the model and associated data are not publicly available. Our empirical analysis suggests that the trained CellCycleNet model provides more accurate cell cycle stage labels than the SVM trained on DAPI intensity and nuclear area features. Specifically, CellCycleNet achieves an area under the receiver operating characteristic curve (AUROC) of 0.95 on the epifluorescence data and 0.94 on the confocal data (Fig 2B and C). This represents an improvement of 0.07–0.08 relative to the SVM. As an alternative performance measure, we also computed the classification accuracy at the selected decision threshold (Fig 2D). By this measure, CellCycleNet also outperforms the SVM, increasing the accuracy by 9.2% on the epifluorescence dataset and 7.1% on the confocal dataset (Fig 2D). To compare the nature of the models' mistakes, we plotted correct and incorrect classifications on a scatterplot of integrated DAPI intensity and nuclear volume (Fig 2E–G). We found that both models made broadly distributed errors with no clear bias (Fig 2E and G). There were 20 shared mistakes in common; CellCycleNet correctly called 38 cases where the SVM failed and the SVM had 20 correct calls where CellCycleNet failed (Fig 2F).

One challenge associated with deep neural network models is their black-box nature, which can make it difficult to understand why a particular model makes a given prediction. To address this difficulty, various saliency methods have been developed (43, 44 *Preprint*, 45, 46), which aim to identify features of the input data that

are most relevant to a given model's output. Accordingly, we applied one of the most commonly used methods, integrated gradients (46), from the Captum model interpretability library (47 *Preprint*), to the set of 8 representative images selected by Apricot (39) (confocal dataset shown in Fig S3 and epiflourescent dataset shown in Fig S4). The spatial patterns identified by integrated gradients suggest that CellCycleNet is focusing on DAPI bright, heterochromatic foci within the nucleus, though we cannot rule out that nuclear size is still considered by the neural network. Future investigation will be needed to determine whether the model recognizes specific organizational states of the underlying genomic regions, the high local density of the DAPI signal at these sites, or other factors.

## Using 3D information improves cell cycle estimation

CellCycleNet and the SVM differ in two primary ways: first, the SVM learns from 2D whereas CellCycleNet learns from 3D images; and second, CellCycleNet uses a CNN architecture and the SVM does not. To isolate the effect of that first difference on performance, we performed an intermediate experiment, in which we trained an SVM model using features extracted from 3D images versus features extracted from 2D projections of the same images. The first SVM model, consistent with Narotamo et al (34), uses two features extracted from 2D maximum-intensity projection images: 2D

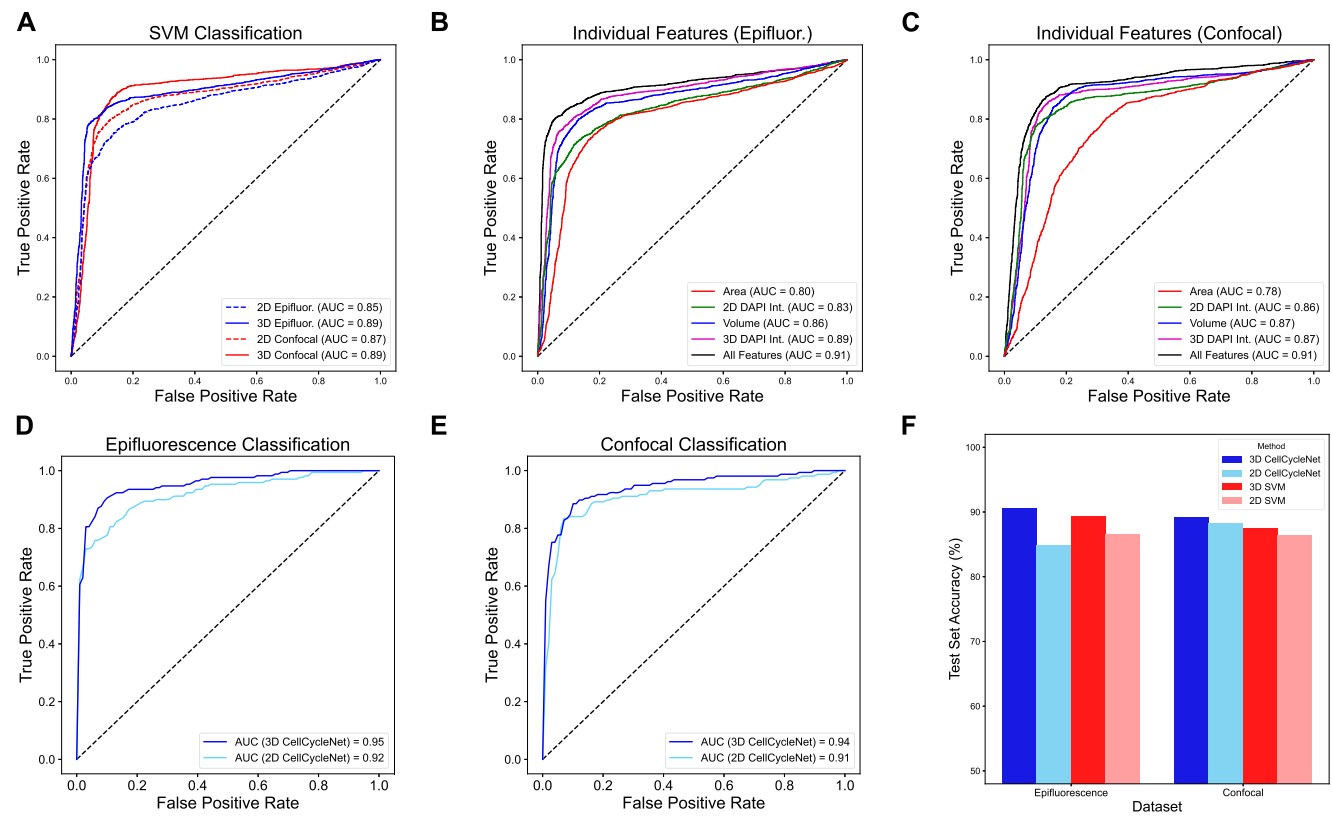

**Figure 3. Using 3D information improves cell cycle estimation for both the SVMs and CellCycleNet.**
**(A)** ROC curves on two image datasets using SVM models. **(B)** ROC curves on confocal image dataset using each individual feature and with four features together for the SVM. **(C)** ROC curves on epifluorescence image dataset using each individual feature and with four features for the SVM. **(D)** ROC curves on epifluorescence dataset using 2D and 3D CellCycleNet. **(D, E)** Same as (D) on the confocal dataset. **(F)** Test accuracy comparison for both 3D and 2D CellCycleNet and SVM models.

integrated DAPI intensity and area. The second SVM model uses two corresponding features derived from the full 3D images: 3D integrated DAPI intensity and volume. All SVM models were implemented in Python with scikit-learn (48). We found that in all cases the RBF kernel achieved the best performance. The regularization and gamma parameters were tuned on a per-model basis using nested cross-validation with scikit-learn's GridSearchCV object (48). To account for differences in magnitude, all features were scaled to a mean of 0 and SD of 1 using z-score normalization. Scaling features prevents a single feature from dominating the decision function, which we found to improve model performance.

Comparing the performance of the 2D and 3D SVM models on the test set images suggests the value of incorporating 3D image information. First, the SVM model integrating 3D features exhibited markedly enhanced classification performance, achieving an AUROC of 0.89 on both the epifluorescence dataset and the confocal dataset (Fig 3A). These values represent a substantial increase, with the 3D model outperforming its 2D counterpart by 4% and 2% in terms of AUROC for the epifluorescence and confocal datasets, respectively. In addition, we also measured the predictive ability of each individual feature (Fig 3B and C) and observed that 3D DAPI integrated intensity is the most predictive single feature among the four. Integrating all four features into a single SVM model, both 2D and 3D, yielded further improvement in AUROC compared with the 3D SVM model (from 0.89 in the epifluorescence dataset and 0.87 in the confocal dataset to 0.91

in both datasets; Fig 3B and C). We next tested if additional 2D features beyond DAPI intensity and nuclear area could further enhance the SVM's performance. We included a wider range of morphology features from the scikit-image package (49): 2D DAPI integrated intensity, nuclear area, convex nuclear area, bounding box area, equivalent diameter area, extent, maximum Feret diameter, and solidity. Though the inclusion of these additional features did increase the performance of the 2D SVM, it still underperformed compared to CellCycleNet on both image datasets (Fig S5A–D).

To get a fairer baseline for comparison beyond SVMs, we also trained a 2D version of CellCycleNet. The 3D version of CellCycleNet outperforms 2D CellCycleNet (Fig 3D–F), and the 2D CellCycleNet outperforms the state-of-the-art 2D SVM (34) based on AUC metric and test accuracy. We observed that a 3D SVM outperforms 2D CellCycleNet in the epifluorescence image dataset (Fig 3F). This observation helps validate that, despite being a more archaic implementation of ML, SVMs can still outperform a model of deep learning architecture.

Lastly, we wanted to test whether CellCycleNet could be repurposed to tackle a more challenging problem: predicting a cell's position along the cell cycle continuum rather than a discrete cell cycle phase. Given that cell cycle progression is inherently continuous rather than categorical, a regression-based approach may provide a more fine-grained perspective on how individual cells transition through different stages. However, this task is considerably more difficult than classification, since the model

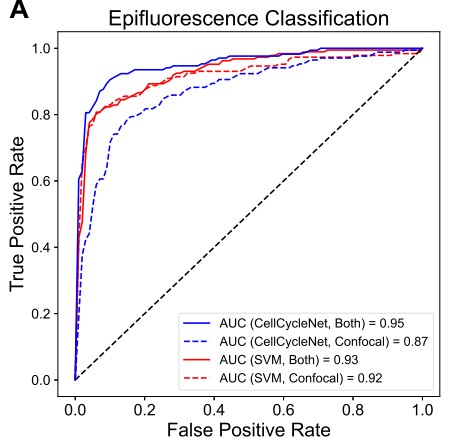

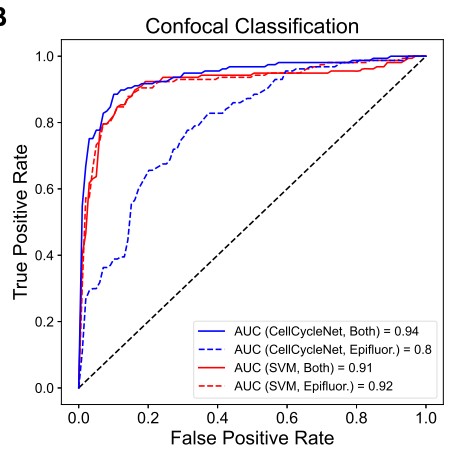

**Figure 4. CellCycleNet does not generalize well across different microscope platforms but improves once trained on both image datasets.**
**(A)** ROC curves on epifluorescence test data.
**(B)** ROC curves on confocal test data.

would need to learn subtle variations in the input features that correspond to gradual cell cycle changes. In the absence of live-cell tracking data explicitly marking a cell's temporal position, we leveraged the ratio of immunofluorescence-boosted Fucci2a red fluorescent protein (mCherry; referred to as RFP) to a green fluorescent protein (mVenus; referred to as GFP) intensities to approximate progression through the cell cycle. Specifically, we modeled a cell's position along the cycle as an angular coordinate, calculated as the angle $\theta$ formed by the line connecting the origin (0,0) to each cell's (RFP, GFP) intensity coordinate in fluorescence space (Fig S6A). This angle encodes the relative balance between RFP and GFP signals: lower values correspond to G1-phase cells while higher values correspond to cells progressing through S and G2 phases.

We adapted CellCycleNet for a regression task by replacing the classification head with a single node equipped with a linear activation function in the final output layer. This modification allowed the model to produce a continuous-valued prediction of the angle rather than a discrete label. We trained this modified version of CellCycleNet using the same confocal dataset as the original classifier, but with the angular position as the target variable. Model performance was evaluated on a held-out test set, where it achieved a root mean squared error of 18.4 and a Pearson correlation coefficient (R) of 0.792 (Spearman: 0.748) (Fig S6B). These results suggest that the model captured meaningful variation in cell cycle position despite the increased difficulty of the regression task. To directly compare performance of the regression model to the original CellCycleNet, we binarized the labeled angle at the threshold where RFP and GFP intensities are equal and used this binary classification to calculate the AUROC value. A well-performing model should be able to learn this distinction reasonably well as in a direct binary classification task. However, its performance may be slightly lower due to the inherent tradeoff between classification accuracy and regression-based fitness. Indeed, we obtained AUROC of 0.92 (Fig S6C), which is slightly lower than the AUROC of 0.94 achieved by the original CellCycleNet classifier. These results indicate that the adapted CellCycleNet successfully captures proxy cell cycle dynamics and could be leveraged for fine-grained temporal ordering or phase progression analysis in single-cell imaging studies.

## CellCycleNet's performance improves when trained on both image datasets

After confirming that CellCycleNet accurately infers cell cycles within each image dataset, and because we acquired images from two different image modalities, we next investigated whether a model trained on data from one platform could generalize well to data from another. We evaluated CellCycleNet's generalizability in both directions: training on epifluorescence data and predicting on confocal data, and vice versa. We found that, for both directions, CellCycleNet trained on one platform performed poorly when tested on the other (Fig 4A and B).

This observation prompted us to test whether training CellCycleNet on both image datasets could match the performance of CellCycleNet trained on the two datasets independently. We found that training on both types of images resulted in the most accurate predictions, achieving an AUROC of 0.95 on the epifluorescence dataset and 0.94 on the confocal dataset (Fig 4). This performance was even better than models trained exclusively on one type of image (Fig 4A and B). In contrast, training an SVM model on the combined data set led to negligible changes in performance (AUROC increased by 0.01 on the epifluorescence dataset and decreased by 0.01 on the confocal dataset). This observation underscores the power of the CNN to learn features intrinsic to cell cycle phasing that generalize across microscope platforms.

## Discussion

We developed a machine learning model, CellCycleNet, for predicting cell cycle stages from 3D images of DAPI-stained nuclei. CellCycleNet is fine-tuned from a pre-trained 3D U-Net (42) using images from two widely used microscope modalities. This model automatically extracts critical 3D features necessary for prediction, eliminating the need for manual selection of morphological features. Our empirical evaluation of the model suggests two useful conclusions. First, incorporating 3D information into either an SVM or a deep neural network greatly improves cell cycle classification accuracy. Current state-of-the-art approaches rely on 2D max-

projected image data from 3D data, which removes information between planes that contain useful features for prediction. Second, a deep neural network outperforms an SVM, particularly when trained on data from two different imaging platforms. CellCycleNet is therefore a more suitable architecture to achieve generalizability across image modalities for 2D and 3D image datasets. Our study provides completely new benchmarking resource data and models, freely available to the community, which we anticipate will be useful for the future development of phenotype classification models.

CellCycleNet's current implementation does have some limitations. As with all machine learning models, users need to consider how well their data matches CellCycleNet's training data and make adjustments accordingly. For example, CellCycleNet was trained on mouse 3T3 fibroblast cells containing a Fucci2a transgene (5). Other "workhorse" cell lines of varied shapes and sizes may be out of distribution for CellCycleNet. To address this, we provide software tools so that users with their own cell line and ground-truth labels can further fine-tune CellCycleNet to achieve more accurate cell cycle stage classification. On a related note, CellCycleNet may be sensitive to 3D segmentation performance. Because each cell line may have different nuclear shapes and sizes, users may need to adjust their segmentation methodology to achieve robust results with CellCycleNet. We encourage users to explore a variety of cell segmentation solutions, such as CellPose (36, 37, 38 Preprint), Stardist (50), CellProfiler (51, 52), or others reviewed by Hollandi et al (53). We also rely on end users to provide their own data quality control steps, as low-quality cells were not included in CellCycleNet's training dataset. Lastly, CellCycleNet currently supports only two cell cycle labels: G1 and G2/S. This limitation is due to the resolving power of the Fucci2a transgene. Ongoing efforts are improving Fucci technology to better resolve three classes (G1, S, and G2), such as with the Fucci(CA) cassette (4). Future work can utilize these next-generation Fucci sensors for improved classification power. Other cell states, like G1 arrest or quiescence, may be inaccessible from in vitro contexts due to transgene silencing. Since Fucci signals are cumulative, a regression task would be informative to pinpoint the precise cell cycle position of each DAPI-stained nucleus. We tested this by performing a regression task and achieved a similar AUROC score as CellCycleNet's original binary classification task. However, it is important to point out the limitation that the angle-based prediction is only a proxy and not a substitute for ground truth. Additionally, because our model was trained on stained, fixed-cell data, its predictions may not translate perfectly to live-cell imaging scenarios. Future work could explore whether incorporating additional fluorescence markers or time-lapse imaging data could further refine the model's ability to track continuous cell cycle progression.

Though we observed that using 3D features from images instead of 2D features was the best for improving classification performance, there are practical considerations that may still make a 2D approach preferable to 3D in some settings. In high-throughput imaging contexts (as in Roukos et al (32)), speed is prioritized to boost cell numbers. As a result, only a few planes are captured per cell in Z, which is insufficient for a true 3D analysis. In this case, users may favor a 2D approach that sacrifices performance and generalizability for an increased number of imaged cells and

reduced cost. CellCycleNet has 2D and 3D functionality to support both modalities.

We anticipate that CellCycleNet will be most useful for experimentalists who work with mouse fibroblast cells for a wide array of applications. Cell cycle progression is a key metric for evaluating cell responses to pharmacological agents, pathogenic stimuli, and in the study of gene mutations. Though there are many ways to measure the cell cycle, these methods are often complex and time-consuming. In contrast, DAPI- staining of fixed cells is fast and is already a part of many image analysis-based workflows. Using CellCycleNet will allow access to information from the same data "for free"—that is, with minimal to no additional hands-on experimenter effort. Additionally, because cell cycle inference uses the DAPI channel that serves as a counterstain, two or more channels are open to supporting additional targets in downstream assays, such as those provided by immunofluorescence or fluorescence in situ hybridization assays. Beyond technical considerations, using CellCycleNet in lieu of classic cell cycle labeling methods will be robust to side effects—both known and unknown—that can confound interpretation of experimental results. For example, BrdU pulses are inherently toxic and have been shown to increase cell death and repress differentiation in neural progenitor cells (2, 54).

We have released all our image data, trained models, and software as a community resource accessible to experimentalists. Lastly, we believe that our key finding—that 3D information significantly improves machine learning model performance—will inform future work in ML-based bioimage analysis beyond cell cycle profiling.

# Materials and Methods

### Cell culture

The Fucci (Fluorescent Ubiquitination-based Cell Cycle Indicator) 3T3 Fucci2a reporter cell line was a gracious gift from Richard Mort (5) and maintained in complete 3T3 complete cell culture media (high glucose 4.5 g/l DMEM with 2-mM L-glutamine, 10% FBS, and 1% pen/strep antibiotic) in the presence of 100–200 $\mu$g/ml hygromycin selection antibiotic. Prior to the imaging experiments, 7 × 10$^4$ cells were seeded in a 35-mm dish. The next day, cells were fixed in 4% PFA in 1xPBS for 10 min, followed by three washes in 1x PBS for 1 min each prior to immunofluorescence staining.

### Immunofluorescence signal amplification

We performed immunofluorescence staining targeting endogenous Fucci2a mVenus and mCherry to boost signal and effectively reduce exposure time. Following cell fixation, cells were permeabilized in 1x PBS containing 0.1% Triton X-100 for 10 min, followed by three 5-min rinses in 1x PBS. Next, cells were blocked with BlockAid Blocking Solution (cat. no. B10710; Thermo Fisher Scientific) containing 0.05% Tween-20 for 30 min at room temperature. Cells were then stained with a primary antibody solution overnight at 4C: 3% BlockAid Blocking Solution in 1xPBS with 1:100 concentration of both

primary goat anti-GFP (cat. No. GPCA-GFP; Encor Biotechnologies) and rabbit anti-RFP (cat. No. 600-401-379; Rockland) antibodies. After three 5-min rinses in 1x PBS, cells underwent secondary antibody staining for 30 min with the following solution: 1:1,000 concentration of donkey anti-goat AF488 (cat. No. A32814; Thermo Fisher Scientific) and donkey anti-rabbit AF555 (cat. no. A32794; Thermo Fisher Scientific) and 143 $\mu$M DAPI (cat. no. D1306; Thermo Fisher Scientific) in 1x PBS. After, cells underwent three five-minute rinses in 1x PBS before covered in SlowFade Gold Antifade Mountant (cat. no. S36936; Thermo Fisher Scientific). Stained cells were stored in the dark at 4C until image data acquisition.

### Epifluorescence dataset acquisition

The epifluorescence image dataset was acquired using a Keyence BZ-X800 fluorescence microscope. All images were acquired using the Keyence Plan Apochromatic 40x objective lens (model PZ-PA40, PlanApo NA 0.95/0.17 mm default). A tiled acquisition was obtained over ~5 mm$^2$ of the cell culture plate, at a fixed z range of 35 $\mu$m, z-step of 0.5 $\mu$m, and optical sectioning width of 10. 100% excitation light (metal halide lamp) was used with no binning and monochromatic camera with a digital zoom of x1.0. Each image consists of three fluorescent channels: nuclear stain DAPI (using the DAPI filter cube; model OP-87762, 360/40 nm excitation filter, 400 nm longpass dichroic mirror, 460/50 nm emission filter; 1/1.2s exposure time), the mVenus-hGeminin-AF488 signal (via the GFP filter cube; model OP-87763, excitation wavelength 470/40 nm, emission wavelength 525/50 nm, dichroic mirror wavelength 495 nm; 1/2.3s exposure time), and the mCherry-hCdt1-AF555 signal (via the Cy3/TRITC filter cube; CHROMA model 49004, excitation wavelength 545/25 nm, emission wavelength 605/70 nm, dichroic mirror wavelength 565 nm; 1/7.5s exposure time). An integrated CCD camera was used in 14-bit mode, without binning and with a physical pixel size of 7.549 $\mu$m, resulting in an effective pixel size of 188.7 nm while imaging at 40x magnification. The final dataset used for analysis and model training consists of 19,170 total images over 90 tiled z-stacks (71 images per z-stack) and three channels; each image is 1,440 x 1,920 pixels. The voxel size for each image in the dataset was 188*188*500 nm (X*Y*Z).

### Confocal dataset acquisition

The confocal image dataset was acquired using a Nikon Ti2 inverted fluorescence microscope with a Yokogawa CSU-W1 SoRa spinning disk unit. A tiled acquisition was obtained over ~21 mm$^2$ of the cell culture plate, at a fixed z range of 30 $\mu$m and a z-step of 0.2 $\mu$m. Each image consists of three fluorescent channels, using excitation light emitted at 30 percent maximal intensity from 405, 488, and 561 nm lasers housed in a Nikon LUNF 405/488/561/640NM 1F commercial launch. The nuclear stain DAPI was excited by the 405 nm laser line at 300-ms exposure; the mVenus-hGeminin-AF488 signal was excited by the 488 nm laser line at 300-ms exposure; the mCherry-hCdt1-AF555 signal was excited by the 561 nm laser line at 300-ms exposure. Laser excitation was conveyed through a single-mode optical fiber fed into the CSU-W1 SoRa unit and directed through a microlens array and "SoRa" disc containing 50 $\mu$m pinholes. A Nikon Plan Apochromat Lambda 60X air objective lens (NA 0.95/0.17 mm)

was used to project the excitation and collect the emission light. Emission light was relayed by a 1x lens, through the pinhole disc, after which it was spectrally separated by a quad bandpass dichroic mirror (Semrock Di01-T405/488/568/647-13x15x0.5) and then filtered by one of three single bandpass filters (DAPI: Chroma ET455/50M; mVenus-hGeminin-AF488: Chroma ET525/36M; mCherry-hCdt1-AF555:Chroma ET605/50M). Emission light was then focused by a 1x relay lens onto an Andor Sona 4.2B-11 camera with a physical pixel size of 11 $\mu$m, resulting in an effective pixel size of 110 nm. The Andor Sona 4.2B-11 camera was operated in 16-bit mode with rolling shutter. The final dataset used for analysis and model training consists of 625 tiled z-stacks (150 images per z-stack) and 3 channels; each image is 1,024 x 1,024 pixels. The voxel size for each image in the dataset was 182*182*200 nm (X*Y*Z).

### 3D segmentation

To quantify the features of single nuclei, we performed 3D instance segmentation with Cellpose (36). Specifically, for the epifluorescence dataset, we used Cellpose 2.0 out-of-the-box using the "CP" model (36, 37). For the confocal dataset, we used the Cellpose 3.0 package with human-in-the-loop fine-tuning of the "cyto3" model (38 Preprint). For each tile in our dataset, we extract the nuclear channel (stained with DAPI) and downsample the image by a factor of 2 in the X and Y dimensions. We found that downsampling improves the segmentation results when using Cellpose's pre-trained models with no fine-tuning. We removed poor segmentation by ignoring objects contacting the image boundary and segmentation with a solidity (ratio of convex hull to total mask volume) of < 0.9.

### Image preprocessing

We implemented several preprocessing steps and applied them to the imaging data before creating the training/validation/testing datasets. First, we used the 3D instance segmentation mask to generate single-nucleus images centered on the nucleus' centroid and padded to a size of 150px*150px*90 (X*Y*Z). Next, each image in the epifluorescence dataset was downsampled by a factor of 2 in the X and Y dimensions to mitigate the data's anisotropy. Confocal images were downsampled by a factor of 2 in the X, Y, and Z dimensions to more closely match the isotropic voxel size of the epifluorescence images. Then, each image was normalized independently using mean normalization. To achieve this, we first calculated the median values of all nonzero pixels for each image in the training dataset. The validation and test datasets were then normalized by dividing the pixel values of each image by the median of medians of images in the training dataset. The final preprocessed images for both datasets were then padded to a shape of 75px*75px*90 (X*Y*Z).

### Data split and augmentation

After each image was preprocessed, we generated the training/validation/testing datasets using a 70%/20%/10% split for each dataset, respectively. We used a fixed random seed during splitting to ensure consistency across different training/evaluation sessions. Data augmentation was used to expand the training set and

improve model generalization. Specifically, a random rotation in the range (0°, 359°) about the Z axis was applied with equal probability.

### Quantification of nuclear features

Each image in the dataset was processed using an analysis pipeline developed in Python. First, each Z-series for each channel in a given tile of the dataset is loaded into memory as a three-dimensional numpy array with the Python package tifffile ([55]). All images consisting of overlap from the image tiling procedure were cropped to ensure there were no duplicate objects. To extract features of interest from the data, we performed 3D instance segmentation on the DAPI channel (as described above) ([36]) to generate masks corresponding to individual nuclei. Each segmentation mask was filtered to remove low-quality segmentation results. Specifically, we removed partial segmentations which contacted the image border, and removed objects considered doublets. To quantify segmentation doublets, we calculated the concavity of the object using the regionprops() function from scikit-image ([49]), and removed objects with a value < 0.9 of the "solidity" property. We then used the filtered segmentation mask to quantify features of interest from each segmented nucleus and store the features in a pandas DataFrame. Features that were calculated are as follows: aggregate volumetric fluorescent intensity in each channel, nuclear volume, lengths of the major and minor axes of each nucleus, and metadata for each object such as the parent tile and object ID. We then stored images of each object's 405/DAPI channel, cropped to a standardized 75px*75px*90 (X*Y*Z) volume.

### Cell cycle ground truth

Ground-truth labels (G1 or S/G2) were generated with reference to the methodology described by Narotamo et al ([34]). Prior to assigning labels, each segmented object was passed through several filters to remove nuclei that did not express the Fucci2a fluorescent proteins, had abnormal sizes, or were transitioning between cell cycle phases. Objects were excluded from our analysis if they met any of the following criteria, where R = mCherry-hCdt1 intensity, G = mVenus-hGeminin intensity, V = number of voxels in segmentation mask, and $\mu$ and $\sigma$ are the mean and SD of each variable, respectively. Values for $\mu_R$ and $\mu_G$ were normalized before the ratio filter was applied to account for differences in fluorescent intensity distributions between channels.

Low-Intensity Filter:
$\mu_R$ < 1, 500 or $\mu_G$ < 2, 200 (Epifluorescence).
$\mu_R$ < 700 or $\mu_G$ < 2, 000 (Confocal).
Ratio Filter:
0.9 < $\mu_G/\mu_R$ < 1.1.
Volume Filter:
$|V - \mu_V| > 2.5\sigma_V$

Of the 5,331 segmented objects in the epifluorescence dataset, 1,255 met one of the above criteria and were filtered out. The remaining 4,076 objects were labeled as G1 if $R > G$ or S/G2 if $R < G$, resulting in 2,203 objects labeled G1 and 1,873 objects labeled S/G2. Of the 5,638 segmented objects in the confocal dataset, 1,912 met one of the above criteria and were filtered out. Labels were

assigned to the remaining 3,726 objects with the previously described method, resulting in 2,096 objects labeled G1 and 1,630 objects labeled S/G2. In both datasets, the majority of filtered objects were excluded due to low fluorescent protein signals.

### CellCycleNet

CellCycleNet is based on a 3D CNN (https://github.com/wolny/pytorch-3dunet) that was trained to perform semantic segmentation of nuclei in light-sheet images ([42]). To retool this model for binary classification, we used the first four convolutional blocks for feature extraction, and we appended four additional layers in total to perform the classification task: one 3DAdaptiveMaxPooling layer with an output size of (1,1,1), one Flatten layer with *start dim* = 1, one Dropout layer with probability 0.5, and one Linear layer with 256 input features and 1 output features. The resulting model contains 35 layers and a total of 1,757,267 parameters. CellCycleNet is trained using binary cross-entropy as the loss function, and each model is trained for 800 epochs, with the checkpoint showing the highest validation accuracy being selected. All models were trained using a batch size of 4, Adam optimizer with an initial learning rate of 1e-5 and weight decay of 1e-1.

A 2D version of CellCycleNet was developed based on the original 3D architecture. The number and order of layers are identical, with the only difference being that the filters are reduced to two dimensions. The model consists of 35 layers and a total of 388,547 parameters. The 2D CellCycleNet was trained for the same number of epochs and with the same hyperparameters as the 3D version. While the 3D CellCycleNet used pre-trained weights, the 2D model was initialized with random weights, because pre-trained weights were not available for the 2D architecture.

## Data Availability

We developed an image processing pipeline with the goal of training an image classification model capable of predicting the cell cycle state from a volumetric imaging dataset of the Fucci2a cells, cultured and fixed in 35-mm plates. All code was developed in Python using the Anaconda package manager to create astandardized analysis environment. Our CellCycleNet is available at https://github.com/Noble-Lab/CellCycleNet. Image data was deposited to the Image Data Resource (https://idr.openmicroscopy.org) under accession number idr0167 (projects 3351, 3352, and 3353).

## Supplementary Information

## Acknowledgements

We thank Dr. Richard Mort and Dr. Jennifer Kong for their gracious gift of the Fucci2a knock-in cell line and 3T3 fibroblast parental cell line, respectively.

EK Nichols thanks all faculty at the Deep Learning for Microscopy Image Analysis Course (2023) at the MBL for their excellent instruction. We thank members of the Noble, Beliveau, and Shendure labs for helpful discussion. This work is supported by NIH award UM1 HG011531. EK Nichols was supported by the Washington Research Foundation Postdoctoral Fellowship. G Li's research is supported by University of Washington eScience Institute.

## Author Contributions

G Li: conceptualization, formal analysis, supervision, investigation, methodology, and writing—original draft, review, and editing.
EK Nichols: conceptualization, data curation, supervision, investigation, and writing—original draft, review, and editing.
VE Browning: data curation, formal analysis, investigation, visualization, methodology, and writing—review and editing.
NJ Longhi: data curation, software, formal analysis, investigation, visualization, and writing—review and editing.
M Sanchez-Forman: software, formal analysis, and validation.
CK Camplisson: software.
BJ Beliveau: conceptualization, resources, supervision, funding acquisition, project administration, and writing—review and editing.
WS Noble: conceptualization, resources, supervision, funding acquisition, project administration, and writing—review and editing.

## Conflict of Interest Statement

The authors declare that they have no conflict of interest.

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
