## [Reviewer comments · Life Science Alliance]

Life Science Alliance

Predicting cell cycle stage from 3D single-cell nuclear-stained images

Gang Li, Eva Nichols, Valentino Browning, Nicolas Longhi, Madison Sanchez-Forman, Conor Camplisson, Brian Beliveau, and William Noble

DOI: <https://doi.org/10.26508/lsa.202403067>

Corresponding author(s): William Noble, University of Washington and Brian Beliveau, University of Washington

Review Timeline:

Submission Date:	2024-09-27
Editorial Decision:	2024-11-25
Revision Received:	2025-03-04
Editorial Decision:	2025-03-06
Revision Received:	2025-03-16
Accepted:	2025-03-17

Transaction Report:

November 25, 2024

Re: Life Science Alliance manuscript #LSA-2024-03067-T

Assistant Professor William Stafford Noble
University of Washington
Genome Sciences
Health Sciences Center, Box 357730
1705 NE Pacific Street
Seattle, WA 98109

Dear Dr. Noble,

Thank you for submitting your manuscript entitled "Predicting cell cycle stage from 3D single-cell nuclear-stained images" to Life Science Alliance. The manuscript was assessed by expert reviewers, whose comments are appended to this letter. We invite you to submit a revised manuscript addressing the Reviewer comments.

Thank you for this interesting contribution to Life Science Alliance. We are looking forward to receiving your revised manuscript.

Sincerely,

B. MANUSCRIPT ORGANIZATION AND FORMATTING:

Reviewer #1 (Comments to the Authors (Required)):

The paper describes the use of a 3D convolutional neural network (3D CNN) to classify the cell cycle position using just the fluorescent nuclear stain (DAPI) images. Ground truth is determined using the Fucci2a cell cycle reporter system as ground truth and the results from the 3D CNN compared with 2D and 3D SVM models.

The authors generate two 3D image dataset using both widefield epifluorescence and spinning disc confocal with different pixel resolutions and these datasets will be made available. This will be a useful resource to the community especially those developing 3D CNNs for phenotype classification.

The 3D CNN is a fairly basic CNN but shows good performance versus very simple SVM models which are using very limited number of features so this is unsurprising. This is also a fairly simple binary classification problem i.e. is the cell in G1 or S/G2 using a DNA marker which discriminates these two phases by a doubling in intensity.

I feel that while the 3D dataset is a great resource the authors have missed an opportunity to be a little more ambitious in terms of trying to pin down more precisely the cell cycle position. Bearing in mind other studies in the literature of cell cycle prediction from 3D images using deep learning, for example

Cell cycle position in 3D spheroid images were reported by
Eismann et. al. *J Cell Sci.* 2020 Jun 1;133(11)

Cell cycle from the 3D structure of chromatin
Wu et al. *Commun Biol* 7, 923 (2024).

or indeed the use of simple 2D images to do label free cell cycle determination or identify mitotic phases etc. then I find the 2 class problem a little simplistic ?

Could the authors use regression to focus in on a more specific cell cycle position since they are using the Fucci2a cell cycle reporter system which could be used to pin-point cell cycle position for example to a percentage of the way through cell cycle ?

Also the authors do not really identify why the 3D network is better than 2D. The 2D SVM models use only a couple of simple features e.g. nuclear area and DAPI intensity which are classic cell cycle position indicators but if others were included e.g. nuclear granularity would this improve accuracy for 2D so the comparison is fairer ?

Also the author suggest the spatial patterns identified by integrated gradients suggests that CellCycleNet is focusing on DAPI foci within the nucleus which may be entirely correct but the integrated gradients plots do not rule out the networking using the nuclear size etc. ?

In summary I believe the dataset is a useful resource and the authors could use this dataset to ask more challenging questions to prove conclusively that the 3D imaging route is the way to go for cell cycle determination.

Reviewer #2 (Comments to the Authors (Required)):

This goal of the work described in this manuscript was the development of a machine learning tool that could accurately classify images of cell stained only with a DNA marker. To facilitate this, the authors produced a dataset of 3D fluorescence microscope images of cells stained with a DNA marker and cell cycle phase indicators and have made them publicly available. A straightforward 3D convolutional neural network was then trained. The accuracy of this network was compared to SVMs trained (both previously and in this study) using basic image features extracted from 2D images. Significant improvement was obtained. The software is made publicly available through github.

The methods and results are clearly described and the public availability of both the image dataset and the software are welcome. The approach is straightforward and does not break any new ground. It is restricted to two classes and does not estimate position of each cell/nucleus within the cell cycle (e.g., early/late G1, mid S). The significance of the manuscript's contribution would appear to rest on estimating its utility to the community, something that is difficult to assess.

It is worth noting that the dataset was extensively filtered before use in training and testing (remove approximately 25% of the cells), leading to some concern that the results would not reflect the accuracy that would be obtained in other studies and whether 2D methods might be more robust.

Reviewer #3 (Comments to the Authors (Required)):

The manuscript presents an application of 3D deep learning for cell cycle stage classification, achieving improved accuracy over some other machine learning methods. While the results are promising, there are areas where the study's claims and impact could be strengthened. Below is the review. Strengths: The manuscript demonstrates that integrating 3D features into models improves the accuracy of cell cycle staging over traditional 2D approaches. It highlights significant performance improvements in terms of AUC. The paper is clearly written.

Limitations: The novelty is limited since it only used commonly found 3D CNN architecture. SVM is an old method to compare here, and not a state-of-the-art model for similar problems. Moreover, a 3D SVM is compared with 3D CycleNet. The authors are encouraged to train a state-of-the-art 2D CycleNet style model to perform the same. Authors are encouraged to discuss and include potentially similar paper as some related works. What are the main differences with those and how exactly this particular paper pushes the boundary of the domain. The list is not exhaustive, but I hope it inspires the author to perform a rigorous literature review.

1. scHiCyclePred: a deep learning framework for predicting cell cycle phases from single-cell Hi-C data using multi-scale interaction information 2. https://github.com/AllenCell/image_classifier_3d 3. Cell Cycle Stage Classification Using Phase Imaging with Computational Specificity.

This part of the paper needs to be clearer "Though we observed that using 3D features from images instead of 2D features was the best for improving classification performance, this does potentially cause a tradeoff for users when it comes to imagedata acquisition. Producing well-sampled 3D data takes more time on the microscope, which limits high-throughput data acquisition and becomes more costly. In some cases, users may opt to adopt a 2D approach that sacrifices performance and generalizability for an increased number of imaged cells and reduced cost."

We thank the editor and editorial team for handling our manuscript and the reviewers for their thorough reading of the manuscript and constructive comments. Below, please find our responses to the reviewer's comments and, where applicable, a summary of changes made to the manuscript.

Indented **blue font** indicates our written response and **green font** indicates edited manuscript text.

Reviewer #1:

The paper describes the use of a 3D convolutional neural network (3D CNN) to classify the cell cycle position using just the fluorescent nuclear stain (DAPI) images. Ground truth is determined using the Fucci2a cell cycle reporter system as ground truth and the results from the 3D CNN compared with 2D and 3D SVM models. The authors generate two 3D image dataset using both widefield epifluorescence and spinning disc confocal with different pixel resolutions and these datasets will be made available. This will be a useful resource to the community especially those developing 3D CNNs for phenotype classification. The 3D CNN is a fairly basic CNN but shows good performance versus very simple SVM models which are using very limited number of features so this is unsurprising. This is also a fairly simple binary classification problem i.e. is the cell in G1 or S/G2 using a DNA marker which discriminates these two phases by a doubling in intensity.

Response: We thank the Reviewer for their careful consideration and review of our work, and their enthusiasm regarding the potential value of our data as a community resource.

I feel that while the 3D dataset is a great resource the authors have missed an opportunity to be a little more ambitious in terms of try to pin down more precisely the cell cycle position. Bearing in mind other studies in the literature of cell cycle prediction from 3D images using deep learning, for example Cell cycle position in 3D spheroid images were reported by Eismann et. al. J Cell Sci. 2020 Jun 1;133(11) Cell cycle from the 3D structure of chromatin Wu et al. Commun Biol 7, 923 (2024). or indeed the use of simple 2D images to do label free cell cycle determination or identify mitotic phases etc. then I find the 2 class problem a little simplistic ? Could the authors use regression to focus in on a more specific cell cycle position since they are using the Fucci2a cell cycle reporter system which could be used to pin-point cell cycle position for example to a percentage of the way through cell cycle ?

Response: We thank the Reviewer for suggesting additional references and sharing their thoughts on our relatively narrow two-class problem. We first expanded our literature search to incorporate additional references for inferring cell cycle stages from different modalities.

We edited the manuscript in the following places to reflect these changes:

- **Page 2 Lines 4-9, expanding range and discussion of literature search:** "Many methods have previously been developed for cell cycle staging across multiple data modalities, including single-cell HiC data [50, 51], single-cell RNA-sequencing [52, 53, 54] and microscopy data. In this work, we focused on cell cycle staging using microscopy image data. These methods are specialized for brightfield or phase contrast images [3, 11, 20, 49, 55], light-sheet microscopy [48] or live cell tracking [46, 18, 27, 22, 21, 1, 17, 37, 28, 4, 43]. We further narrowed our scope to include only studies that had similar input data and cell cycle staging tasks as this study: fixed interphase cells stained with DAPI and imaged by fluorescence microscopy (**Table 1**)."

We further performed a regression analysis using the 3D DAPI image as input to predict each cell's position within the cell cycle. Specifically, we modeled the angle in a 2D scatter plot of RFP and GFP signals, defined as the angle between the x-axis and the line connecting the origin (0,0) to each cell's coordinates (RFP, GFP). This angle serves as a proxy for cell cycle position along the cell cycle trajectory. We modified the final layer of our neural network to

include a single node with a linear activation function. The model achieved a root mean squared error (RMSE) of 18.4 and a Pearson correlation (R) of 0.792 (Spearman: 0.748) on the test dataset. Additionally, we used the predicted angle as a binary classifier to distinguish G1 from G2/M-phase cells, obtaining an AUROC of 0.918, which is slightly less than the CellCycleNet model's 0.941. This approach provides a valuable framework for quantitatively tracking cell cycle position and opens the door for further improvements in predictive modeling and application in single-cell biology.

Figure S6. Regression model for precise cell cycle positioning. The cell cycle position is represented as the angle between the x-axis and the line connecting the origin (0,0) to each cell's RFP-GFP signal coordinates. **(A)** Scatter plot of RFP-GFP signal coordinates, colored by the angle. **(B)** Scatter plot comparing the predicted angle versus the ground truth angle in the test dataset, demonstrating the model's accuracy. **(C)** ROC curves comparing the performance of the 3D regression model and the 3D CellCycleNet classifier, where a decision boundary of 45 degrees is used to classify G1 vs. G2/M cells.

We edited the manuscript in the following places to reflect these changes:

- **Page 2 Lines 32-34, motivating the choice of binary classifier:** "To have a straightforward training objective and to aid in comparisons to the prior art [36], we chose a binary classification model for testing."
- **Page 2 Lines 28-30, additional explanation of the Fucci2a sensor and its limits:** "The hCdt1 protein accumulates during G1, which degrades at the start of S phase. Meanwhile, hGeminin accumulates during S phase and is degraded at the start of G1 [28]; G1 and G2/S phases of the cell cycle can therefore be faithfully resolved with the Fucci2a transgene."
- **Page 8 Lines 1-25, discussing the regression experiment results:** "Lastly, we wanted to test whether CellCycleNet could be repurposed to tackle a more challenging problem: predicting a cell's position along the cell cycle continuum rather than a discrete cell cycle phase. Given that cell cycle progression is inherently continuous rather than categorical, a regression-based approach may provide a more fine-grained perspective on how individual cells transition through different stages. However, this task is considerably more difficult than classification, since the model would need to learn subtle variations in the input features that correspond to gradual cell cycle changes. In the absence of live cell tracking data explicitly marking a cell's temporal position, we leveraged the ratio of Fucci2a red fluorescent protein (RFP) to green fluorescent protein (GFP) intensities to approximate progression through the cell cycle. Specifically, we modeled a cell's position along the cycle as an angular coordinate, calculated as the angle θ formed by the line connecting the origin (0,0) to each cell's (RFP, GFP) intensity coordinate in fluorescence space (**Supplemental Figure S6A**). This angle encodes the relative balance between RFP and GFP signals: lower values correspond to G1-phase cells while higher values correspond to cells progressing through S and G2 phases."

We adapted CellCycleNet for a regression task by replacing the classification head with a single node equipped with a linear activation function in the final output layer. This modification allowed the model to produce a continuous-valued prediction of the angle rather than a discrete label. We trained this modified version of CellCycleNet using the same confocal dataset as the original classifier, but with the angular position as the target variable. Model performance was evaluated on a held-out test set, where it achieved a root mean squared error (RMSE) of 18.4 and a Pearson correlation coefficient (R) of 0.792 (Spearman: 0.748) (**Supplemental Figure S6B**). These results suggest that the model captured meaningful variation in cell cycle position despite the increased difficulty of the regression task. To directly compare performance of the regression model to the original CellCycleNet, We binarized the labeled angle at the threshold where RFP and GFP intensities are equal and used this binary classification to calculate the AUROC value. A well-performing model should be able to learn this distinction reasonably well as in a direct binary classification task. However, its performance may be slightly lower due to the inherent tradeoff between classification accuracy and regression-based fitness. Indeed, we obtained AUROC of 0.918 (**Supplemental Figure S6C**), which is slightly lower than the AUROC of 0.941 achieved by the original CellCycleNet classifier. These results indicate that the adapted CellCycleNet successfully captures proxy cell cycle dynamics and could be leveraged for fine-grained temporal ordering or phase progression analysis in single-cell imaging studies.”

- **Page 9, Lines 27-34, discussing the regression experiment results:** “Other cell states, like G1 arrest or quiescence, may be inaccessible from *in vitro* contexts due to transgene silencing. Since Fucci signals are cumulative, a regression task would be informative to pinpoint the precise cell cycle position of each DAPI-stained nucleus. We tested this by performing a regression task and achieved a similar AUROC score as CellCycleNet’s original binary classification task. However, it is important to point out the limitation that the angle-based prediction is only a proxy and not a substitute for ground truth. Additionally, because our model was trained on stained, fixed-cell data, its predictions may not translate perfectly to live-cell imaging scenarios. Future work could explore whether incorporating additional fluorescence markers or time-lapse imaging data could further refine the model’s ability to track continuous cell cycle progression.”

Also the authors do not really identify why the 3D network is better than 2D.

Response: We thank the Reviewer for their comment. We have now incorporated a 2D version of CellCycleNet as a baseline model, and we demonstrate that the 3D CellCycleNet outperforms 2D CellCycleNet in both image datasets. The results empirically confirm that the 3D network performs better than 2D in both datasets. The primary advantage of a 3D network lies in its ability to process the richer information present in 3D cell images compared to 2D images. As demonstrated in **Figure 3B-C**, each 3D morphological feature exhibits greater predictive power than its corresponding 2D counterpart. This finer and more detailed 3D morphological information cannot be effectively captured by a 2D machine learning model.

D **E**
Figure 3. Using 3D information improves cell cycle estimation. **D.** ROC curves on epifluorescence dataset using 2D and 3D CellCycleNet. **E** same as **D** on the confocal dataset.

We edited the manuscript in the following places to reflect these changes:

- Page 7 Lines 22-26, Figure 3D-E, addressing 2D vs 3D CellCycleNet:**
 “In order to get a fairer baseline for comparison, we also trained a 2D version of CellCycleNet. The 3D version of CellCycleNet outperforms 2D CellCycleNet (**Figure 3D-F**), and the 2D CellCycleNet outperforms the state-of-the-art 2D SVM [31] based on AUC metric and test accuracy. We observed that a 3D SVM outperforms 2D CellCycleNet on the epifluorescence image dataset (**Figure 3F**). This observation helps validate that, despite being a more archaic implementation of ML, SVMs can still outperform a model of deep learning architecture.”
- Page 7, Lines 5-10, new Figure 3 caption:**
 “**Figure 3:** Using 3D information improves cell cycle estimation for both the SVMs and CellCycleNet. **A.** ROC curves on two image datasets using SVM models. **B.** ROC curves on confocal image dataset using each individual feature with using four features together with SVM. **C.** ROC curves on epifluorescence image dataset using each individual feature and with four features for the SVM. **D.** ROC curves on epifluorescence dataset using 2D and 3D CellCycleNet. **E** Same as **D** on the confocal dataset. **F.** Test accuracy comparison for both 3D and 2D CellCycleNet and SVM models.”

The 2D SVM models use only a couple of simple features e.g. nuclear area and DAPI intensity which are classic cell cycle position indicators but if others were included e.g. nuclear granularity would this improve accuracy for 2D so the comparison is fairer ?

Response: We thank the Reviewer for bringing up this important point. We ran an experiment using an SVM that had access to a wider range of 2D features (a total of 8 features, including 2D DAPI integrated intensity, nuclear area, convex nuclear area, bounding box area, equivalent diameter area, extent, maximum Feret diameter, and solidity). The results show that these features lead to the improved performance of the 2D SVM, but the model still underperforms 3D CellCycleNet on both image datasets.

Figure S5. 3D CellCycleNet outperforms the improved 2D SVM models with all 2D morphology features (8 in total) on both image datasets.

We edited the manuscript in the following place to reflect this change:

- Page 5 Lines 27-33, addressing simplistic SVM models by adding more features:** :
 “To test whether providing additional 2D features beyond DAPI intensity and nuclear area enhances the SVM’s performance, we included a wider range of morphology features from the scikit-image package (total of eight: including 2D DAPI integrated intensity, nuclear area, convex nuclear area, bounding box area, equivalent diameter area, extent, maximum Feret diameter, and solidity). Though inclusion of these features did increase the performance of the 2D SVM, it still underperformed compared to 3D CellCycleNet on both image datasets (**Figure S5**).”

Also the author suggest the spatial patterns identified by integrated gradients suggests that CellCycleNet is focusing on DAPI foci within the nucleus which may be entirely correct but the integrated gradients plots do not rule out the networking using the nuclear size etc. ?

Response: We thank the Reviewer for bringing up this nuanced point. We agree that although the integrated gradients highlight DNA foci within the nucleus (and not the border of nuclei), this analysis does not rule out the possibility that CellCycleNet is also using nuclear size information. Integrated gradients focuses on relative importance, so while DAPI-bright foci may show strong contributions, nuclear size might still play a lesser role.

We edited the manuscript in the following place to reflect this change:

- Page 5 Lines 39-41, addressing integrated gradients:**
 “The spatial patterns identified by integrated gradients suggests that CellCycleNet is focusing on DAPI bright, heterochromatic foci within the nucleus, though we cannot rule out that nuclear size is still considered by the neural network.”

In summary I believe the dataset is an useful resource and the authors could use this dataset to ask more challenging questions to prove conclusively that the 3D imaging route is the way to go for cell cycle determination.

Response: After discussion with some members of the community—who could be potential end users of our method—we realized that there are situations where a 3D imaging route would not be a practical choice for them, even if it does provide superior results. For example, in a high-

throughput imaging context, it is infeasible to acquire thousands of well-sampled 3D images due to data size constraints and acquisition time. Instead, minimally 3D (“2.5D”) or 2D images are typically acquired (Reference #36). In these instances, a 2D CellCycleNet classifier or an improved SVM could be sufficiently accurate for their purposes. Ultimately, we chose to leave it up to the investigator whether the extra boost in performance that 3D information brings is worth the additional up-front cost of acquiring it.

We edited the manuscript in the following place to reflect this change:

- **Page 9 Lines 36-40, discussing recommendations for imaging:**
“Though we observed that using 3D features from images instead of 2D features was the best for improving classification performance, there are practical considerations that may still make a 2D approach preferable to 3D in some settings. In high-throughput imaging contexts (as in *Roukos et al.* [36]), speed is prioritized to boost cell numbers. As a result, only a few planes are captured per cell in Z, which is insufficient for a true 3D analysis. In this case, users may favor a 2D approach that sacrifices performance and generalizability for an increased number of imaged cells and reduced cost. CellCycleNet has 2D and 3D functionality to support both modalities.”

Reviewer #2:

This goal of the work described in this manuscript was the development of a machine learning tool that could accurately classify images of cell stained only with a DNA marker. To facilitate this, the authors produced a dataset of 3D fluorescence microscope images of cells stained with a DNA marker and cell cycle phase indicators and have made them publicly available. A straightforward 3D convolutional neural network was then trained. The accuracy of this network was compared to SVMs trained (both previously and in this study) using basic image features extracted from 2D images. Significant improvement was obtained. The software is made publicly available through github. The methods and results are clearly described and the public availability of both the image dataset and the software are welcome.

Response: We thank the Reviewer for their careful consideration and review of our work.

The approach is straightforward and does not break any new ground.

Response: We thank the Reviewer for raising this concern about novelty. Our major take-away is that using 3D information, irrespective of model architecture, outperforms 2D information alone. Currently, to the best of our knowledge, nobody has yet incorporated 3D information into their model training for this task on similar data, even when the raw data is natively 3D (References #30, 31). We thought this was surprising. We believe that our work therefore brings a valuable insight that may be useful for others as they develop models for other tasks involving bioimage data, and our software provides a practical tool for end users interested in cell cycle classification from their own imaging data.

It is restricted to two classes and does not estimate position of each cell/nucleus within the cell cycle (e.g., early/late G1, mid S).

Response: We thank the Reviewer for sharing their thoughts on our relatively narrow two-class problem. To address the concern about our two-class approach, we performed a regression analysis using the 3D DAPI image as input to predict each cell’s position within the cell cycle.

We further performed a regression analysis using the 3D DAPI image as input to predict each cell’s position within the cell cycle. Specifically, we modeled the angle in a 2D scatter plot of RFP and GFP signals, defined as the angle between the x-axis and the line connecting the origin (0,0) to each cell’s coordinates (RFP, GFP). This angle serves as a proxy for cell cycle

position along the cell cycle trajectory. We modified the final layer of our neural network to include a single node with a linear activation function. The model achieved a root mean squared error (RMSE) of 18.4 and a Pearson correlation (R) of 0.792 (Spearman: 0.748) on the test dataset. Additionally, we used the predicted angle as a binary classifier to distinguish G1 from G2/M-phase cells, obtaining an AUROC of 0.918, which is slightly less than the CellCycleNet model's 0.941. This approach provides a valuable framework for quantitatively tracking cell cycle position and opens the door for further improvements in predictive modeling and application in single-cell biology.

Figure S6. Regression model for precise cell cycle positioning. The cell cycle position is represented as the angle between the x-axis and the line connecting the origin (0,0) to each cell's RFP-GFP signal coordinates. **(A)** Scatter plot of RFP-GFP signal coordinates, colored by the angle. **(B)** Scatter plot comparing the predicted angle versus the ground truth angle in the test dataset, demonstrating the model's accuracy. **(C)** ROC curves comparing the performance of the 3D regression model and the 3D CellCycleNet classifier, where a decision boundary of 45 degrees is used to classify G1 vs. G2/M cells.

We edited the manuscript in the following places to reflect these changes:

- **Page 2 Lines 32-34, motivating the choice of binary classifier:** "To have a straightforward training objective and to aid in comparisons to the prior art [36], we chose a binary classification model for testing."
- **Page 2 Lines 28-30, additional explanation of the Fucci2a sensor and its limits:** "The hCdt1 protein accumulates during G1, which degrades at the start of S phase. Meanwhile, hGeminin accumulates during S phase and is degraded at the start of G1 [28]; G1 and G2/S phases of the cell cycle can therefore be faithfully resolved with the Fucci2a transgene."
- **Page 8 Lines 1-25, discussing the regression experiment results:** "Lastly, we wanted to test whether CellCycleNet could be repurposed to tackle a more challenging problem: predicting a cell's position along the cell cycle continuum rather than a discrete cell cycle phase. Given that cell cycle progression is inherently continuous rather than categorical, a regression-based approach may provide a more fine-grained perspective on how individual cells transition through different stages. However, this task is considerably more difficult than classification, since the model would need to learn subtle variations in the input features that correspond to gradual cell cycle changes. In the absence of live cell tracking data explicitly marking a cell's temporal position, we leveraged the ratio of Fucci2a red fluorescent protein (RFP) to green fluorescent protein (GFP) intensities to approximate progression through the cell cycle. Specifically, we modeled a cell's position along the cycle as an angular coordinate, calculated as the angle θ formed by the line connecting the origin (0,0) to each cell's (RFP, GFP) intensity coordinate in fluorescence space (**Supplemental Figure S6A**). This angle

encodes the relative balance between RFP and GFP signals: lower values correspond to G1-phase cells while higher values correspond to cells progressing through S and G2 phases.

We adapted CellCycleNet for a regression task by replacing the classification head with a single node equipped with a linear activation function in the final output layer. This modification allowed the model to produce a continuous-valued prediction of the angle rather than a discrete label. We trained this modified version of CellCycleNet using the same confocal dataset as the original classifier, but with the angular position as the target variable. Model performance was evaluated on a held-out test set, where it achieved a root mean squared error (RMSE) of 18.4 and a Pearson correlation coefficient (R) of 0.792 (Spearman: 0.748) (**Supplemental Figure S6B**). These results suggest that the model captured meaningful variation in cell cycle position despite the increased difficulty of the regression task. To directly compare performance of the regression model to the original CellCycleNet, We binarized the labeled angle at the threshold where RFP and GFP intensities are equal and used this binary classification to calculate the AUROC value. A well-performing model should be able to learn this distinction reasonably well as in a direct binary classification task. However, its performance may be slightly lower due to the inherent tradeoff between classification accuracy and regression-based fitness. Indeed, we obtained AUROC of 0.918 (**Supplemental Figure S6C**), which is slightly lower than the AUROC of 0.941 achieved by the original CellCycleNet classifier. These results indicate that the adapted CellCycleNet successfully captures proxy cell cycle dynamics and could be leveraged for fine-grained temporal ordering or phase progression analysis in single-cell imaging studies.”

- **Page 9, Lines 27-34, discussing the regression experiment results:** “Other cell states, like G1 arrest or quiescence, may be inaccessible from *in vitro* contexts due to transgene silencing. Since Fucci signals are cumulative, a regression task would be informative to pinpoint the precise cell cycle position of each DAPI-stained nucleus. We tested this by performing a regression task and achieved a similar AUROC score as CellCycleNet’s original binary classification task. However, it is important to point out the limitation that the angle-based prediction is only a proxy and not a substitute for ground truth. Additionally, because our model was trained on stained, fixed-cell data, its predictions may not translate perfectly to live-cell imaging scenarios. Future work could explore whether incorporating additional fluorescence markers or time-lapse imaging data could further refine the model’s ability to track continuous cell cycle progression.”

The significance of the manuscript’s contribution would appear to rest on estimating its utility to the community, something that is difficult to assess.

Response: We thank the Reviewer for acknowledging the potential resource value of our work, and we are sorry that we have not made the significance of our work more clear in our manuscript. Our manuscript’s contribution is fourfold: first, our proposed CellCycleNet outperforms the state-of-the-art method; second, we provide completely new benchmarking resource data and models available for everyone; third, we have shown that, for both SVM and CellCycleNet, using 3D information provides more accurate prediction than 2D; last but not least, in our most recent revised version, we also trained and released a 2D version of CellCycleNet, which can be useful for people with 2D images.

We edited the manuscript in the following places to reflect these changes:

- **Page 7 Lines 22-26, Figure 3D-E, addressing 2D vs 3D CellCycleNet:** “In order to get a fairer baseline for comparison, we also trained a 2D version of CellCycleNet. The 3D version of CellCycleNet outperforms 2D CellCycleNet (**Figure 3D-F**), and the 2D CellCycleNet outperforms the state-of-the-art 2D SVM [31] based on AUC metric and test accuracy. We observed that a 3D SVM outperforms 2D CellCycleNet on the epifluorescence

image dataset (**Figure 3F**). This observation helps validate that, despite being a more archaic implementation of ML, SVMs can still outperform a model of deep learning architecture.”

- Page 7, Lines 5-10, new Figure 3 caption:**
“Figure 3: Using 3D information improves cell cycle estimation for both the SVMs and CellCycleNet. **A.** ROC curves on two image datasets using SVM models. **B.** ROC curves on confocal image dataset using each individual feature with using four features together with SVM. **C.** ROC curves on epifluorescence image dataset using each individual feature and with four features for the SVM. **D.** ROC curves on epifluorescence dataset using 2D and 3D CellCycleNet. **E** Same as **D** on the confocal dataset. **F.** Test accuracy comparison for both 3D and 2D CellCycleNet and SVM models.”
- Page 10 Lines 9-12, discussing publicly available image data resource:**
 “CellCycleNet is therefore a more suitable architecture to achieve generalizability across image modalities for 2D and 3D image datasets. Our study provides completely new benchmarking resource data and models, freely available to the community, which we anticipate will be useful for the future development of phenotype classification models.”
- Page 10 Lines 30-35, discussing imaging recommendations and choice of 2D or 3D mode:**
 “Though we observed that using 3D features from images instead of 2D features was the best for improving classification performance, there are practical considerations that may still make a 2D approach preferable to 3D in some settings. In high-throughput imaging contexts (as in Roukos et al. [36]), speed is prioritized to boost cell numbers. As a result, only a few planes are captured per cell in Z, which is insufficient for a true 3D analysis. In this case, users may favor a 2D approach that sacrifices performance and generalizability for an increased number of imaged cells and reduced cost. CellCycleNet has 2D and 3D functionality to support both modalities.”

It is worth noting that the dataset was extensively filtered before use in training and testing (remove approximately 25% of the cells), leading to some concern that the results would not reflect the accuracy that would be obtained in other studies and whether 2D methods might be more robust.

Response: We apologize for not being more upfront about the sources of data loss in our pipeline for each data type. We are following the original SVM baseline paper from Narotamo et al. 2021 [31] to remove nuclei for various reasons. Among the ~25% of cells that are filtered, 76-88% cannot be rescued because they don't have FUCCI signals. 12-24% of nuclei are along the border, too large, or too small.

Nuclei number (proportion)	Epifluorescence Dataset	Confocal Dataset
No FUCCI signal cells	950 (17.8%)	1685 (29.9%)
Border cells between classes	202 (3.8%)	164 (2.9%)
Too large or too small	103 (1.9%)	63 (1.1%)
Passed cells	4076 (76.5%)	3726 (66.1%)

We omitted this data to improve the validity of our performance measures. Inclusion of partial or poorly segmented nuclear images and/or inconclusive ground truth labels during model training and evaluation would negatively impact performance. We designed CellCycleNet to be used downstream of the nuclear segmentation and quality control process, which is a part of most, if not all, 2D and 3D single-cell image analysis workflows.

We edited the manuscript in the following places to reflect these changes:

- **Page 4 Lines 18-24, additional description of image data filtration:**
“To ensure that only high-quality cells were used in model training, we removed improperly segmented cells (5.7% of cells in the epifluorescence dataset and 4% in the confocal dataset) or cells that did not express the Fucci2a transgene (17.8% of cells in the epifluorescence dataset and 29.9% of cells in the confocal dataset). Improper segmentations include doublet cells, cell debris, and truncated cells at tile borders. About 65–75% of cells in each dataset passed quality control. After quantifying the relative RFP and GFP fluorescent intensity per cell, we omitted cells of indistinguishable class along the S-phase border, due to the limited resolution power of the Fucci2a transgene.”
- **Page 10, Lines 20-23, discussion regarding CellCycleNet post-segmentation:**
“We encourage users to explore a variety of cell segmentation solutions, such as CellPose [41, 33, 40], Stardist [44], CellProfiler [5, 26], or others reviewed by Hollandi et al. [19]. We also rely on end users to provide their own data quality control steps, as low-quality cells were not included in CellCycleNet’s training dataset.”

Reviewer #3:

Remarks to the Author:

The manuscript presents an application of 3D deep learning for cell cycle stage classification, achieving improved accuracy over some other machine learning methods. While the results are promising, there are areas where the study's claims and impact could be strengthened. Below is the review.

Strengths: The manuscript demonstrates that integrating 3D features into models improves the accuracy of cell cycle staging over traditional 2D approaches. It highlights significant performance improvements in terms of AUC. The paper is clearly written.

Response: We thank the Reviewer for their careful review and thoughtful critiques.

Limitations: The novelty is limited since it only used commonly found 3D CNN architecture. SVM is an old method to compare here, and not a state-of-the-art model for similar problems.

Response: While we agree with the reviewer that SVMs are relatively old, we do not agree that it does not represent the state of the art in this particular area. Indeed, in our literature search for the specific task of cell cycle prediction from DAPI-stained cell nuclear images, the best-performing and most recent model uses an SVM (Narotamo et al. 2021 [31]). While the same first authors have produced a 2D CNN model in an older work [30], they did not release their model and code, which makes reproducibility difficult. These observations motivated our choice of the SVM as the state-of-the-art comparison.

We edited the manuscript in the following place to reflect this change:

- **Page 3 Lines 17-24, additional description of image data filtration:**

“We chose the SVM from Narotamo et al. [31] as the state-of-the-art method because it was the most recent ML method for this task with similar input data (**Table 1**). Though Narotamo et al. [30] have also developed a 2D CNN in older work, the model and associated data are not publicly available. Our empirical analysis suggests that the trained CellCycleNet model provides more accurate cell cycle stage labels than the SVM trained on DAPI intensity and nuclear area features.”

Based on the reviewer’s feedback, we added a new 2D version of CellCycleNet. We found that the 3D SVM model actually outperforms our 2D CNN model. This result suggests that the SVM is a legitimate baseline to compare against—a deep learning architecture is not always going to perform better (new **Figure 3F**).

We edited the manuscript in the following place to reflect this change:

- Page 7 Lines 22-26, Figure 3D-E, addressing 2D vs 3D CellCycleNet:**
 “In order to get a fairer baseline for comparison, we also trained a 2D version of CellCycleNet. The 3D version of CellCycleNet outperforms 2D CellCycleNet (**Figure 3D-F**), and the 2D CellCycleNet outperforms the state-of-the-art 2D SVM [31] based on AUC metric and test accuracy. We observed that a 3D SVM outperforms 2D CellCycleNet on the epifluorescence image dataset (**Figure 3F**). This observation helps validate that, despite being a more archaic implementation of ML, SVMs can still outperform a model of deep learning architecture.”
- Page 7, Lines 5-10, new Figure 3 caption:**
 “**Figure 3:** Using 3D information improves cell cycle estimation for both the SVMs and CellCycleNet. **A.** ROC curves on two image datasets using SVM models. **B.** ROC curves on confocal image dataset using each individual feature with using four features together with SVM. **C.** ROC curves on epifluorescence image dataset using each individual feature and with four features for the SVM. **D.** ROC curves on epifluorescence dataset using 2D and 3D CellCycleNet. **E** Same as **D** on the confocal dataset. **F.** Test accuracy comparison for both 3D and 2D CellCycleNet and SVM models.”

Last but not least, we chose the current architecture partially because we have a relatively limited training dataset. We adopt transfer learning to accelerate the convergence from models trained on similar images. In the future, if there are other sophisticated architectures with model weights trained on similar images, we could fine-tune in a similar fashion for cell cycle classification tasks.

We edited the manuscript in the following place to reflect this change:

- Page 4 Lines 13-14, regarding model architecture and transfer learning:**
 “The model is a 3D convolutional neural network that processes 3D images through various operations, including multiple layers of convolution and max pooling. This transfer learning step helped to accelerate convergence.”

Moreover, a 3D SVM is compared with 3D CycleNet. The authors are encouraged to train a state-of-the-art 2D CycleNet style model to perform the same.

Response: We thank the Reviewer for this great suggestion. We have added a 2D version of CellCycleNet to our comparison. As shown in revised **Figure 2**, the 3D version of CellCycleNet outperforms 2D CellCycleNet.

We edited the manuscript in the following places to reflect these changes:

- Page 7 Lines 22-26, Figure 3D-E, addressing 2D vs 3D CellCycleNet:**
 “In order to get a fairer baseline for comparison, we also trained a 2D version of CellCycleNet. The 3D version of CellCycleNet outperforms 2D CellCycleNet (**Figure 3D-F**), and the 2D CellCycleNet outperforms the state-of-the-art 2D SVM [31] based on AUC metric and test accuracy. We observed that a 3D SVM outperforms 2D CellCycleNet on the epifluorescence image dataset (**Figure 3F**). This observation helps validate that, despite being a more archaic implementation of ML, SVMs can still outperform a model of deep learning architecture.”
- Page 7, Lines 5-10, new Figure 3 caption:**
 “**Figure 3:** Using 3D information improves cell cycle estimation for both the SVMs and CellCycleNet. **A.** ROC curves on two image datasets using SVM models. **B.** ROC curves on confocal image dataset using each individual feature with using four features together with SVM. **C.** ROC curves on epifluorescence image dataset using each individual feature and with four features for the SVM. **D.** ROC curves on epifluorescence dataset using 2D and 3D
 ”

CellCycleNet. **E** Same as **D** on the confocal dataset. **F**. Test accuracy comparison for both 3D and 2D CellCycleNet and SVM models.”

Authors are encouraged to discuss and include potentially similar paper as some related works. What are the main differences with those and how exactly this particular paper pushes the boundary of the domain. The list is not exhaustive, but I hope it inspires the author to perform a rigorous literature review.

1. schiCyclePred: a deep learning framework for predicting cell cycle phases from single-cell Hi-C data using multi-scale interaction information
2. https://github.com/AllenCell/image_classifier_3d3. Cell Cycle Stage Classification Using Phase Imaging with Computational Specificity.

Response: Thank you for providing potentially related works for us to consider. We first expanded our literature search to incorporate additional references for inferring cell cycle stages from different modalities such as schiC and phase imaging.

Though we did an exhaustive literature search at the outset of the study, we did not cite all papers in the interest of brevity. We focused on microscopy-based approaches that take in DAPI-stained nuclear images from in vitro cell culture. In the revised manuscript, we have now added an additional sentence justifying our selection of related work.

We edited the manuscript in the following places to reflect these changes:

- **Page 2 Lines 4-9, expanding range and discussion of literature search:** “Many methods have previously been developed for cell cycle staging across multiple data modalities, including single-cell HiC data [50, 51], single-cell RNA-sequencing [52, 53, 54] and microscopy data. In this work, we focused on cell cycle staging using microscopy image data. These methods are specialized for brightfield or phase contrast images [3, 11, 20, 49, 55], light-sheet microscopy [48] or live cell tracking [46, 18, 27, 22, 21, 1, 17, 37, 28, 4, 43]. We further narrowed our scope to include only studies that had similar input data and cell cycle staging tasks as this study: fixed interphase cells stained with DAPI and imaged by fluorescence microscopy (**Table 1**).”

- **We added the following references:**

[47] N. Zielke and B. A. Edgar. FUCCI sensors: powerful new tools for analysis of cell proliferation. *WIREs Developmental Biology*, 4(5):469–487, 2015. eprint: <https://onlinelibrary.wiley.com/doi/pdf/10.1002/wdev.189>.

[48] B. Eismann, T. G. Krieger, J. Beneke, R. Bulkescher, L. Adam, H. Erfle, C. Herrmann, R. Eils, and C. Conrad. Automated 3d light-sheet screening with high spatiotemporal resolution reveals mitotic phenotypes. *Journal of cell science*, 133(11):jcs245043, 2020.

[49] Y. R. He, S. He, M. E. Kandel, Y. J. Lee, C. Hu, N. Sobh, M. A. Anastasio, and G. Popescu. Cell cycle stage classification using phase imaging with computational specificity. *ACS photonics*, 9(4):1264–1273, 2022.

[50] T. Nagano, Y. Lubling, C. V´arnai, C. Dudley, W. Leung, Y. Baran, N. Mendelson Cohen, S. Wingett, P. Fraser, and A. Tanay. Cell-cycle dynamics of chromosomal organization at single-cell resolution. *Nature*, 547(7661):61–67, 2017.

[51] Y. Wu, Z. Shi, X. Zhou, P. Zhang, X. Yang, J. Ding, and H. Wu. schicyclepred: a deep learning framework for predicting cell cycle phases from single-cell hi-c data using multi-scale interaction information. *Communications Biology*, 7(1):923, 2024.

[52] X. Guo and L. Chen. From g1 to m: a comparative study of methods for identifying cell cycle phases. *Briefings in Bioinformatics*, 25(2):bbad517, 2024.

[53] A. Riba, A. Oravec, M. Durik, S. Jim´enez, V. Alunni, M. Cerciat, M. Jung, C. Keime, W. M. Keyes, and N. Molina. Cell cycle gene regulation dynamics revealed by rna velocity and deep-learning. *Nature communications*, 13(1):2865, 2022.

[54] S. C. Zheng, G. Stein-O’Brien, J. J. Augustin, J. Slosberg, G. A. Carosso, B. Winer, G. Shin, H. T. Bjornsson, L. A. Goff, and K. D. Hansen. Universal prediction of cell-cycle position using transfer learning. *Genome biology*, 23(1):41, 2022.

[55] Leger LA, Leonardi M, Salati A, Naef F, Weigert M. Sequence models for continuous cell cycle stage prediction from brightfield images. *arXiv preprint arXiv:2502.02182*. 2025 Feb 4.

This part of the paper needs to be clearer "Though we observed that using 3D features from images instead of 2D features was the best for improving classification performance, this does potentially cause a tradeoff for users when it comes to image data acquisition. Producing well-sampled 3D data takes more time on the microscope, which limits high-throughput data acquisition and becomes more costly. In some cases, users may opt to adopt a 2D approach that sacrifices performance and generalizability for an increased number of imaged cells and reduced cost."

Response: We thank the Reviewer for this great suggestion. We wanted to be fair and acknowledge that, while incorporating 3D information resulted in the best model performance, it might not be the most practical choice. In our revised manuscript, we also trained and released a 2D version of CellCycleNet (*see above*), which can be used by people who only have 2D images. The 2D version outperforms state-of-the-art SVM models but underperforms 3D CellCycleNet (**Figure 3F**). Ultimately, we chose to leave it up to the investigator whether the extra boost in performance that 3D information brings is worth the additional up-front cost of acquiring it. We have modified our manuscript to clarify the utility of our model.

We edited the manuscript in the following place to reflect this change:

- **Page 9 Lines 36-40, discussing imaging recommendations and choice of 2D or 3D mode:** "Though we observed that using 3D features from images instead of 2D features was the best for improving classification performance, there are practical considerations that may still make a 2D approach preferable to 3D in some settings. In high-throughput imaging contexts (as in *Roukos et al.* [36]), speed is prioritized to boost cell numbers. As a result, only a few planes are captured per cell in Z, which is insufficient for a true 3D analysis. In this case, users may favor a 2D approach that sacrifices performance and generalizability for an increased number of imaged cells and reduced cost. CellCycleNet has 2D and 3D functionality to support both modalities."

March 6, 2025

RE: Life Science Alliance Manuscript #LSA-2024-03067-TR

Dr. William Stafford Noble
University of Washington
Genome Sciences
Health Sciences Center, Box 357730
1705 NE Pacific Street
Seattle, WA 98109

Dear Dr. Noble,

Thank you for submitting your revised manuscript entitled "Predicting cell cycle stage from 3D single-cell nuclear-stained images". We would be happy to publish your paper in Life Science Alliance pending final revisions necessary to meet our formatting guidelines.

- please be sure that the authorship listing and order is correct
- please upload your main and supplementary figures as single files
- please add ORCID ID for the secondary corresponding author- they should have received instructions on how to do so
- please add the X and Bluesky handles of your host institute/organization as well as your own or/and one of the authors in our system
- please upload a clean manuscript file without the colored text
- please add your main, supplementary figure, and table legends to the main manuscript text after the references section
- please remove the figures from the manuscript text and leave them uploaded separately
- the contributions selected for William Stafford Noble and Brian J. Beliveau do not qualify them for authorship. Please either update the contributions in our system and the Author Contributions section of the manuscript or let us know if the authors need to be removed (and added eventually to the acknowledgment section)
- please use the [10 author names, et al.] format in your references (i.e., limit the author names to the first 10)
- please add callouts for Figures 2A,E,F,G; S2; S4 and S5A-D to your main manuscript text

A. FINAL FILES:

B. MANUSCRIPT ORGANIZATION AND FORMATTING:

Sincerely,

March 17, 2025

RE: Life Science Alliance Manuscript #LSA-2024-03067-TRR

Dr. William Stafford Noble
University of Washington
Genome Sciences
Health Sciences Center, Box 357730
1705 NE Pacific Street
Seattle, WA 98109

Dear Dr. Noble,

Thank you for submitting your Methods entitled "Predicting cell cycle stage from 3D single-cell nuclear-stained images". It is a pleasure to let you know that your manuscript is now accepted for publication in Life Science Alliance. Congratulations on this interesting work.

DISTRIBUTION OF MATERIALS:

Again, congratulations on a very nice paper. I hope you found the review process to be constructive and are pleased with how the manuscript was handled editorially. We look forward to future exciting submissions from your lab.

Sincerely,
